# How and Why to Manipulate Your Own Agent: On the Incentives of Users of Learning Agents

**Yoav Kolumbus**
The Hebrew University of Jerusalem
`yoav.kolumbus@mail.huji.ac.il`

**Noam Nisan**
The Hebrew University of Jerusalem
`noam@cs.huji.ac.il`

## Abstract

The usage of automated learning agents is becoming increasingly prevalent in many online economic applications such as online auctions and automated trading. Motivated by such applications, this paper is dedicated to fundamental modeling and analysis of the strategic situations that the *users* of automated learning agents are facing. We consider strategic settings where several users engage in a repeated online interaction, assisted by regret-minimizing learning agents that repeatedly play a "game" on their behalf. We propose to view the outcomes of the agents' dynamics as inducing a "meta-game" between the users. Our main focus is on whether users can benefit in this meta-game from "manipulating" their own agents by misreporting their parameters to them. We define a general framework to model and analyze these strategic interactions between users of learning agents for general games and analyze the equilibria induced between the users in three classes of games. We show that, generally, users have incentives to misreport their parameters to their own agents, and that such strategic user behavior can lead to very different outcomes than those anticipated by standard analysis.

## 1   Introduction

This paper deals with the following common type of scenario: several users engage in some strategic online interaction, where each of them is assisted by a learning agent. A typical example is advertisers that compete for advertising slots over some platform. Typically, each of these advertisers enters his key parameters into some advertiser-facing website, and then this website's "agent" participates on the advertiser's behalf in a sequence of auctions for ad slots. Often, the platform designer provides this agent as its advertiser-facing user interface. In cases where the platform's agent does not optimize sufficiently well for the advertiser (but rather, say, for the auctioneer), one would expect some other company to provide a better (for the advertiser) agent.

The basic model for such scenarios is as follows. There is an underlying $n$-person game that $n$ "software learning agents" will play repeatedly. Some of the parameters of this game, typically the players' utilities, are private to the $n$ "human users," and each of these users enters his key parameters into his own software agent. From this point on, these software agents repeatedly play the game on behalf of their users, where each agent aims to optimize the utility of its owner according to the parameters reported to it by using some learning algorithm. A typical learning algorithm of this type will have at its core some regret-minimization algorithm [15, 47], such as "multiplicative weights" [1], "online gradient descent" [94], or some variant of fictitious play [18, 79], such as "follow the perturbed leader" [45, 51].

On an intuitive level, these agents, playing between themselves, each aiming to maximize its owner's utility, will reach some kind of a game-theoretic equilibrium of the underlying game, and the average utilities of the agents over time will converge to the utilities of this equilibrium. Specifically, while it is known that the dynamics of regret-minimization algorithms may fail to converge to any equilibrium

36th Conference on Neural Information Processing Systems (NeurIPS 2022).

[6, 7, 48, 76], it is also known that the empirical play statistics of no-regret dynamics do approach what is sometimes called a *coarse correlated equilibrium* [15, 47, 92] (CCE). The notion of coarse correlated equilibrium is a generalized weaker notion than the Nash equilibrium,[1] and is essentially a formalization of the no-regret property for each agent. For formal definitions and further discussion and analysis of regret-minimization dynamics, see Appendix A in the Supplementary Material.

While the dynamics of learning agents in repeated games and their convergence properties have been the focus of extensive studies since the early days of game theory [13, 14, 18, 45, 79, 83] and in later works, e.g., [15, 26, 35, 36, 37, 47, 88], our focus is on a different question of analyzing the *incentives of the users* of such agents. In our scenario where (human) users report some of their parameters to their (software) agents, the learning process of the agents depends on the parameters that the users report to them. This motivates the following question: **Given a game and a set of learning agents (one for each user), what parameters should the users report to their own learning agents?**

Since in the reached equilibrium every agent is best-replying to the others' behavior, it would seem obvious that each user would maximize his utility by indeed *reporting his true parameters to his own agent*, thus indeed allowing the agent to optimize for the true utilities. However, as we show, on closer inspection, this is not necessarily true: while a user is guaranteed that his agent best-replies to the others' empirical play (in the sense of having low regret), this empirical play itself of the other agents and the outcome of the joint dynamics are dependent on the behavior of our user's own agent. Thus, the interactions between the agents induce a "meta-game" between the users, in which the users' actions are the parameters that they report to their own agents, and the users' utilities are determined by the long-term empirical average of the agents' joint dynamics.

This is somewhat similar to the situation in classic repeated games: since agents respond to the previous actions of the other players, the "dynamics" do not necessarily give an equilibrium of the underlying single-shot game. In fact, the folk theorem characterizes a wide set of equilibria of the repeated game that can be reached using the correct combination of punishments and rewards that each player uses to affect the others' behavior (see, e.g., [11, 52, 75, 81]). In some sense, the critical aspect here is a "theory of mind" that players have about each other, which enables them to understand how punishments and rewards affect the other players' play in the equilibrium of the repeated game.

By contrast, in our case, regret-minimizing agents have no "theory of mind" since they are each blindly following their own regret-minimization strategy. In particular, a user's agent never aims to punish or reward another agent. A user that aims to influence the outcome of the repeated play between regret-minimizing agents must in some sense have a "theory of mind" of regret-minimizing agents, both of his own agent as well as of the other agents.

In this paper we establish the first steps in the theoretical modeling and analysis of the incentives of users of such automated learning agents in online strategic systems. We analyze three classes of games: dominance-solvable games, Cournot competition games, and opposing-interests games, and demonstrate how in all these different settings, users generally have incentives to manipulate their own learning agents by misreporting their parameters to them, and how equilibria of the users' game can have different properties and outcomes than those of the original underlying game.

Our results underline the importance of considering and analyzing *user incentives*: when a strategic system is accessible to its users through learning agents, its original properties are not necessarily preserved, and even strong notions like strict domination may no longer hold, and so a meta-game analysis is required in order to understand or anticipate the actual outcomes. Our results focus on demonstrating the types of phenomena that can happen due to strategic behavior of users of learning algorithms and on showcasing them in games that are as transparent for analysis and as simple as possible. The framework that we propose is general for any learning dynamics and can be studied in any game – either analytically or by using simulations. In a companion paper [53] we use the model that we propose here to analyze repeated auctions with regret-minimizing agents, and show that these phenomena indeed occur also in the auction setting. That is, counterintuitively, in the (non-truthful) first-price auction users prefer to submit truthful reports to their agents, while in the (dominant-strategy truthful) second-price auction users have incentives to manipulate their own agents by submitting non-truthful reports of their valuations.

---

[1]For games in which there are at most two actions for each player this notion is equivalent to Aumann's correlated equilibrium [2, 3], but for general games it is a weaker and more general notion.

## 2 The Meta-Game Model

We define the "meta-game" between "users" ("players") in terms of the outcomes of the repeated games that their agents play on their behalf. In our definition, all that we formally need from these agents is that they repeatedly interact with each other, where at each point in time each agent's algorithm has observed the past play of all agents in all previous time steps, and then needs to determine its own next action based on this information (and the parameters given by its user). Intuitively, however, we are thinking of agents that aim to maximize some utility for their user, specifically, regret-minimizing agents.

**Definition 1.** *A* user–agent meta-game *has the following ingredients.*

- **Users and Agents***: We have $n$ "human" users, each with his fixed "software" learning agent.*

- **Agent Strategy Spaces** $A_1, ..., A_n$*: The agents "play" against each other for a period of $T$ time steps, where in each step $t = 1, ..., T$ each agent $i$ plays an action $a_i^t \in A_i$ and then gets as feedback the actual play of all the agents $\boldsymbol{a}^t = (a_1^t, ..., a_n^t)$.*

- **User Parameter Spaces** $P_1, ..., P_n$*: Each user $i$ inputs a "declaration" $p_i \in P_i$ into his agent.*

- **Agent Utility Functions** $u_1, ..., u_n$*: The utility function of agent $i$ is $u_i : P_i \times A_1 \times \cdots \times A_n \to \mathcal{R}$. At each time step $t$, each agent $i$ gets a utility of $u_i(p_i, \boldsymbol{a}^t)$, where $p_i$ are the parameters given to it by its user. Our agents aim to maximize this utility.*

- **User True Types** $s_1, ..., s_n$*: Each user has a "true" type $s_i \in P_i$ that describes the true utility of the user. The utility of user $i$ when his true type is $s_i$, and when each user $j$ declares $p_j \in P_j$ to his agent, is $U_i(s_i, p_1, ..., p_n) = (\sum_{t=1}^{T} u_i(s_i, \boldsymbol{a}^t))/T$, where for each $t = 1, ..., T$, $\boldsymbol{a}^t$ is the vector of actions played by the agents at time step $t$, as defined above. If the agents are randomized, then this expression is actually a random variable, and we define $U_i$ to be its expectation.*

- **The User Meta-Game***: Fixing the true parameters $(s_1, ..., s_n)$, we get an $n$-person game between the users, where user $i$'s strategy space is $P_i$ and his utility is $U_i(s_i, \cdot)$.*

This model is formally defined for any game, fixed tuple of learning algorithms, and fixed "horizon" $T$, and for any such the ensuing meta-game can be directly studied by simulations. In order to proceed and theoretically study this meta-game as the horizon $T$ goes to infinity, we would require a setting where it is possible to theoretically analyze the utilities of the resulting dynamics both for the true types (parameters) $s_i$ and for possible deviations $p_i \neq s_i$. The theoretical analysis in the following sections concerns all regret-minimization algorithms, in the "limit" $T \to \infty$, as we will study cases where the agents' game converges to a known CCE. For further discussion on the convergence of regret-minimizing agents, see Appendix A. We note that while our focus is on regret-minimization dynamics, all our results hold also under an alternative interpretation of our model in which the agents reach a CCE of the game with the declared parameters via an arbitrary black-box device.

For the games studied, we will start with basic questions like what is the best reply of a player to another player in the meta-game, and then proceed to more advanced questions, specifically, in which cases is the truth $p_i = s_i$ a best reply in the meta-game, and when this is not the case, what is the Nash equilibrium of the meta-game. All proofs in this paper are deferred to the appendix.

**Definition 2.** *A user–agent meta-game is called* manipulation-free *if the truth-telling declaration profile, i.e., all users declaring $p_i = s_i$, is a Nash equilibrium of the meta-game.*[2]

## 3 Dominance-Solvable Games

The first class of games that we consider are dominance-solvable games. A main interest in the literature in studying these types of games has been as design objectives due to their stable strategic structure, e.g., for voting mechanisms [31, 66] and contract design [5, 44, 80]. Formally, games of this class have a unique pure Nash equilibrium that is also the single CCE, and thus we know that the

---

[2]This definition is for a fixed game that is defined by the true parameters $s_1, ..., s_n$ (but in the context of given parameter spaces $P_1, ..., P_n$). One may also naturally look at the family of games for all possible $s_1, ..., s_n$ and discuss truthfulness in the sense used in mechanism design [71], but we leave this for further follow-up work.

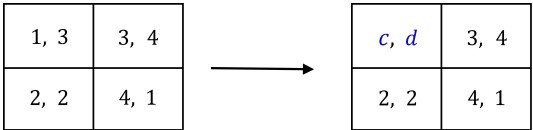

Figure 1: An example of parameter manipulation in a dominance-solvable game. Left: the payoff matrix of the true game. Right: the manipulated payoffs that the players provide to their learning agents. The row player selects the parameter $c$ and the column player selects the parameter $d$.

empirical play statistics of any regret-minimization dynamics will provide, in the limit, the utilities of this equilibrium. For completeness, we give here the definition of a dominance-solvable game.

**Definition 3.** *A game is called* dominance solvable *if there exists an order of iterated elimination of strictly dominated strategies that leads to a single strategy profile (the unique Nash equilibrium).*

We show that even in these strategically simple games, users of learning agents face non-trivial strategic considerations, and, specifically, even users who have a dominant strategy in the game may obtain further gains (beyond their dominant-strategy outcome) by manipulating their own agents.

We begin by demonstrating our agenda on a simple $2 \times 2$ two-person game where one of the players has a dominant strategy. The other player then has a strict best reply, and so the game is dominance-solvable. It is not difficult to see that any dynamics of regret-minimizing learning agents playing such a game will converge to the pure equilibrium, since the agent with the dominant strategy will learn to play only its dominating strategy, regardless of the actions of the second agent, and then the second agent will learn to best-reply to that. Formally, the time average of regret-minimization dynamics must converge to the (unique) Nash equilibrium as it is also the unique CCE. For further details, see Appendix B.

This simple analysis also shows that the other player has no profitable manipulation: since the agent of the player with the dominant strategy can indeed learn to play it whatever the other player does, the other player can do no better than to best-reply to the dominant strategy.

It may also seem intuitive that the player with the dominant strategy can have no profitable manipulation either, but this turns out to be false. Consider the game depicted in Figure 1 (left), in which the row player has a dominant strategy to play the bottom row. The unique pure Nash equilibrium of this game gives the row player utility $u_1 = 2$. We now get to our point where the utility of each player is private to him (at least partially). In our example, suppose that the constants 1 and 3 in the game description are private to the two users, respectively, and that each learning agent gets the value of the parameter ($c$ and $d$, respectively) from their users. After each agent gets its own parameter, the two agents then engage in repeated play. In this play, each agent minimizes regret for its owner *in the game with the parameters that were given to it* (rather than the true parameters that are known only to the user), as shown in Figure 1 (right). If the row player declares instead of the true $c = 1$ a value of, say, $c = 5$, then the (declared) game has a unique mixed Nash equilibrium, which is also its unique CCE [20, 67], where the row player plays the top row with probability $p = 1/2$, and the column player plays the left column with probability $q = 1/4$, giving the row player a utility of $u_1 = 3$. Declaring an even higher value $c \to \infty$ will decrease $q \to 0$, leading to a higher utility of $u_1 \to 3.5$.

Once the row player has manipulated his input, the column player may also beneficially do so. The meta-game does not literally have a Nash equilibrium (as the strategy spaces of the players are a continuum and no continuity of utility is guaranteed), but it has an $\epsilon$-equilibrium[3] for any $\epsilon > 0$: as $c \to \infty$ and $d = 4 - \delta$ with $\delta > 0$ and $\delta \to 0$. In this case, $p \to 1$ and $q \to 0$, leading to an $\epsilon$-equilibrium with utilities $u_1 \to 3$, $u_2 \to 4$.

Notice that the outcome in this $\epsilon$-equilibrium of the meta-game strictly Pareto-dominates the Nash equilibrium of the original game, and thus we may say that the players managed to reach a cooperative outcome. The logic behind this cooperation is that the player with the dominant strategy is given the opportunity to take the point of view of a Stackelberg game where he goes first and the other player best-replies to his strategy. If the dominated strategy is the preferable strategy in the Stackelberg game, as it is in our example, then a manipulation can approach it.

---

[3]In an $\epsilon$-equilibrium no player can gain more than $\epsilon$ by deviating.

This example demonstrates how even in very simple games, users may have incentives to misreport their parameters to their agents, and that even a strong notion like strict domination does not guarantee the stability of truthful declarations. This phenomenon is in fact general for a large class of games:

**Theorem 1.** *In any $n \times m$ game where one of the players has a dominant strategy, if the Stackelberg outcome of the game is different from the truthful Nash equilibrium outcome, then, for a sufficiently wide user parameter space, the game is not manipulation-free, and, specifically, the player with the dominant strategy has an incentive to manipulate his declaration.*

For the special case of $2 \times 2$ games this result holds even when users can manipulate only a single one of their parameters, and for specific subclasses of games it is possible to characterize the equilibria of the meta-game, as in our example above. Additionally, Theorem 1 is in fact even more general and applies not only to games with a dominant strategy, but to any dominance-solvable $n \times m$ game where some player has a Stackelberg value that is higher than his utility in the truthful Nash equilibrium.

# 4  Cournot Competition Games

The second class of games that we consider are the classic Cournot competition games [25, 34, 63] with linear demand functions and linear production costs. These games are contained in a class of games called "socially concave" that were identified by [34], who showed that for games of this class the time-average distribution of any regret-minimization dynamics converges to the unique Nash equilibrium of the game, and so we can confidently analyze the utilities obtained in the meta-game.

We consider a game between two firms that are competing for buyers by controlling the quantity that each of them produces. There is a demand function that specifies the market price for any given total quantity produced. In our case we assume that the demand function is linear; i.e., if the two firms produce quantities $q_1$ and $q_2$ respectively then the market price will be $a - b \cdot (q_1 + q_2)$, where $a$ and $b$ are commonly known positive constants. The private parameter that each firm will have is its production costs, which we also assume are linear; i.e., firm $i$'s cost to produce quantity $q_i$ is exactly $c_i \cdot q_i$, where $0 \leq c_i \leq a$ is privately known to firm $i$. The utility of firm 1 is given by $u_1(q_1, q_2) = q_1 \cdot (a - b \cdot (q_1 + q_2) - c_1)$; similarly, $u_2(q_1, q_2) = q_2 \cdot (a - b \cdot (q_1 + q_2) - c_2)$.

The Nash equilibrium of the game depends on the parameters as follows. If $a + c_2 - 2c_1 > 0$ and $a + c_1 - 2c_2 > 0$, then the Nash equilibrium is $q_1 = \frac{1}{3b}(a + c_2 - 2c_1)$ and $q_2 = \frac{1}{3b}(a + c_1 - 2c_2)$. If $a + c_1 - 2c_2 > 0$ and $c_1 < a$, the Nash equilibrium is $q_1 = \frac{a - c_1}{2b}$ and $q_2 = 0$, and symmetrically, if $a + c_2 - 2c_1 > 0$ and $c_2 < a$, the equilibrium is $q_1 = 0$ and $q_2 = \frac{a - c_2}{2b}$. Otherwise, in the Nash equilibrium both players produce zero.

Thus, there are four parameter regions of interest associated with the four possible types of unique Nash equilibria of the game, as illustrated in Figure 2. The parameter region $A = \{c_1, c_2 | a + c_2 - 2c_1 > 0,\ a + c_1 - 2c_2 > 0,\ c_1 > 0,\ c_2 > 0\}$ is the region where both agents produce positive quantities (the shaded areas in region $A$ in the figure relate to equilibria of the meta-game, as explained below). The parameter regions $B = \{c_1, c_2 | a + c_1 - 2c_2 > 0,\ 0 < c_1 < a,\ c_2 > 0\}$ and $C = \{c_1, c_2 | a + c_2 - 2c_1 > 0,\ 0 < c_2 < a,\ c_1 > 0\}$ are regions where only one player produces a positive quantity. In the remaining region, region $D = \{c_1, c_2 | c_1, c_2 \geq a\}$, both agents produce zero.

As a running example, we will consider the case where $a = b = 1$ and $c_1 = c_2 = 1/2$, for which the standard analysis yields that in equilibrium each player produces the quantity $q_i = 1/6$, the price is thus $2/3$, and the utility of each player is $1/36$.

We now turn to look at the meta-game in which each player reports his production cost to his own agent, and then the agents repeatedly play the declared game. That is, each player $i$ reports a declared cost $0 \leq x_i \leq a$ in the parameter space and then the agents reach the equilibria with the *declared* costs. It turns out that in our example, firm 1's best reply to firm 2's true cost is $x_1 = 3/8$, rather than the truth $c_1 = 1/2$, which increases its utility (calculated, of course, according to the true costs) to $1/32 > 1/36$. That is, the firm under-represents its production costs to its own agent, causing the agent to over-produce. While this over-production by itself hurts our user (the firm), the benefit is that our user – as opposed to its automated learning agent – understands that this aggressive declaration will lead to a reduction in the production of the other firm's agent, making up, and more, for the revenue loss from its own over-production.

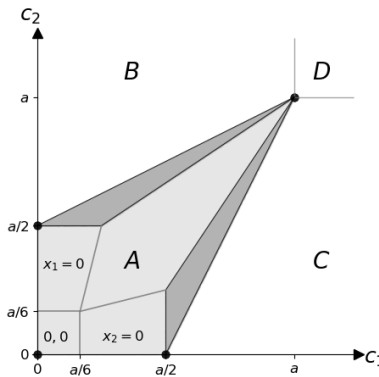

Figure 2: Parameter declaration regions in Cournot competition games.

Analyzing the equilibrium of the meta-game between the users, one obtains that the equilibrium is the declaration profile $x_1 = x_2 = 2/5$, rather than the truthful declarations $c_1 = c_2 = 1/2$. That is, the two players under-represent their production costs to their agents, causing them to over-produce, where each of them produces a quantity $q_i = 1/5$, which is strictly higher than the production of $1/6$ obtained in the equilibrium of the truthful game. In the equilibrium of the meta-game the utility of each of the two players drops to $1/50$, which is significantly less than the original utility of $1/36$. We see that the players are locked here in a sort of a prisoners' dilemma where each of them benefits from a unilateral deviation from the truth, but when they both deviate, they both suffer losses.

Our analysis shows that these comparative statics generalize as long as both players keep producing a non-zero quantity in the meta-game equilibrium. In some cases, which we explicitly describe below, the player with lower costs can under-represent his costs in a sufficiently extreme way so as to drive the other player completely out of the market. In such cases the quantity produced by the player who remains in the market is still larger than the total quantity produced by both players when they play the truth in the meta-game, but his utility increases. We also completely characterize the rather limited set of cases where the meta-game is manipulation-free.

**Theorem 2.** *(1) In any two-player linear Cournot competition with linear production costs, the total quantity produced in the equilibrium of the meta-game is greater than or equal to the total quantity produced when the players play the truth in the meta-game, and the price is thus lower. (2) If both players continue to produce a non-zero quantity in the meta-game equilibrium, then their utilities are less than or equal to their utilities when both play the truth in the meta-game. If, on the other hand, one of the players produces zero in the meta-game equilibrium, then the utility of the producing player is greater than or equal to his utility when both play the truth in the meta-game.*

**Theorem 3.** *In a two-player linear Cournot competition with linear production costs, the meta-game is manipulation-free if and only if either (at least) one of the players produces zero when both play the truth in the meta-game, or both players have zero production costs, $c_i = 0$.*

Figure 2 illustrates the parameter ranges in which the different types of equilibria of the meta-game exist. The dark-shaded areas show the parameter ranges where in the equilibrium of the meta-game the player with the low production cost drives the other competitor out of the market, and the light-shaded area shows the parameter range where both players declare costs lower than their true costs and produce positive quantities in the equilibrium of the meta-game. The regions denoted in the figure by $x_1 = 0$ and $x_2 = 0$ show where players 1 and 2, respectively, declare a cost of zero to their agents in equilibrium, and the region denoted by $0, 0$ is the range where both players declare zero in equilibrium. In the remaining kite-shaped region in region $A$, both players declare positive costs that are less than their true costs. Finally, in regions $B, C$, and $D$, there is no competition and the equilibrium declarations are truthful. For further details, see Appendix C.

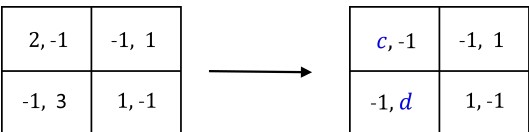

Figure 3: An example of parameter manipulation in an opposing-interests game. Left: the payoff matrix of the true game. Right: the manipulated payoffs that the players provide to their learning agents. The row player selects the parameter $c$ and the column player selects the parameter $d$.

## 5 Opposing-Interests Games

The next class of games that we analyze are games in which there is a single mixed Nash equilibrium and no pure equilibrium. The prototypical example is matching pennies. In these games, often called fully mixed games, the unique Nash equilibrium is also the unique coarse correlated equilibrium [20, 67], and thus the dynamics of regret-minimizing agents will approach this equilibrium. We focus on a subclass called *opposing-interests* games, where the first player gets higher utilities along the main diagonal than he gets along the other diagonal, and the opposite is true for the second player. This subclass of fully mixed games includes many games that are similar to matching pennies, and specifically includes all constant-sum games that are fully mixed.

We begin with an example of the following variant of matching pennies, where one of the utilities for each player is changed from the standard value of 1 to another value, as shown in Figure 3 (left).

A standard analysis shows that the single mixed Nash equilibrium of this game is where the row player plays the top row with probability $p = 2/3$ (and plays the bottom row with probability $1 - p = 1/3$), and the column player plays the left column with probability $q = 2/5$ (and the right column with probability $1 - q = 3/5$). Calculating the utilities of the two players in this equilibrium gives us $u_1 = 1/5$ for the row player, and $u_2 = 1/3$ for the column player. Running a (typical) simulation of multiplicative-weights learning agents that repeatedly play this game against each other, we observe the dynamics shown in Figure 4a. As is well known [7, 48, 76], and as we can clearly see, there is no convergence in the behavior of the agents. However, if we write down the empirical probabilities of play of each of the four combinations of the players' strategies (as shown in Figure 4b) we get (close to) the Nash equilibrium probabilities, as theoretically expected [20].[4]

In our example, suppose that the constants 2 and 3 in the game description are parameters that the two users, respectively, declare to their agents, as shown in Figure 3 (right). Now we ask ourselves, what should the players do in order to maximize their utility? What parameter should, say, the row player report to his agent so as to maximize his expected utility over the whole run of the learning agents? It would seem natural to assume that entering the true value, in our case 2, should be the best possible: after all, the agent is optimizing for the value entered into it, and so the row player should give his agent the correct value to optimize for. However, as we have seen also in other types of games, this intuition is again false. Suppose that the row player reports, e.g., $c = 1$ as his parameter to his learning agent (and suppose that the column player sticks to the truth, $d = 3$). When the two agents now repeatedly play against each other they reach (in the limit empirical distribution sense) the Nash equilibrium of the game with these values of the parameters, which, as one may calculate, is $p = 2/3$, $q = 1/2$. The utility of the row player, whose true value is $c = 2$, in this resulting distribution is $u_1 = 1/3$, which is greater than his "truth-telling" utility of $1/5$, as we have seen above.[5]

---

[4]There are two reasons for not reaching exactly the Nash equilibrium. First, as our simulations are only for a finite number of steps and with a finite step size, the multiplicative-weights algorithm does not fully minimize regret but only nearly so, and thus leads only to a near-equilibrium. Second, as the algorithm is randomized there is an expected stochastic error. These error terms are theoretically of an order of magnitude of $O(1/\sqrt{T})$, where $T$ is the number of rounds, which fits the observed deviations in our simulation for $T = 50,000$ rounds.

[5]This improvement is even more puzzling when we take a closer look: in the Nash equilibrium of the original game, the row player has mixed between his two pure strategies, implying that they both give him the same utility. Thus, the only way that our row agent can improve his utility is by first causing the column player's agent to change its distribution of play, and only then it could take advantage of this change. The column player's agent's behavior, however, does not depend directly on the row player's utilities. The solution is that there is an indirect dependence that goes through the actual play of the row agent. Taking advantage of this indirect dependence obviously requires understanding the "theory of mind" of the column agent's algorithm. In our case, as the row user understands that the agents' dynamics will reach the Nash equilibrium of the *declared game*, he

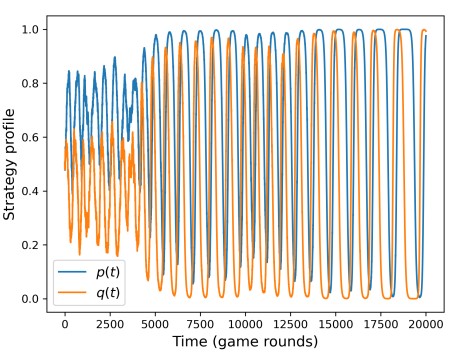

| Nash equilibrium distribution | | |
|---|---|---|
| | $q = 0.4$ | $1 - q = 0.6$ |
| $p = 0.667$ | 0.267 | 0.133 |
| $1 - p = 0.333$ | 0.4 | 0.2 |

| Empirical distribution | | |
|---|---|---|
| | $q = 0.405$ | $1 - q = 0.595$ |
| $p = 0.665$ | 0.270 | 0.135 |
| $1 - p = 0.335$ | 0.395 | 0.200 |

(a) Mixed strategy dynamics    (b) Empirical and Nash equilibrium action distributions

Figure 4: Dynamics and payoffs in the game shown in Figure 3. Left: dynamics of multiplicative-weights agents in the game with the true parameters $c = 2$, $d = 3$. Right: the Nash equilibrium distribution and the empirical action distribution from simulations of multiplicative-weights agents.

Just as the row player can gain by misreporting his utility to his agent, so can the column player. Had the column player declared $d = 1$ (now with the row player declaring the truth $c = 2$), his utility would have increased to $u_2 = 2/5$. Had they both manipulated their "bids" and declared $c = 1, d = 1$ to their respective agents, then both would have benefited relative to telling the truth, getting utilities of $u_1 = 1/4$ and $u_2 = 1/2$.

Next, let us look for an equilibrium of the users' game. We need to determine for which declarations $c, d$ will neither of the two players wishes to unilaterally change his declaration (where the true utilities are still set to $c = 2, d = 3$ in our example). Our analysis shows that the equilibrium declarations in the meta-game, are $c = 3$, $d = 1/3$. The equilibrium of the declared game that is played between the agents in this case is $p = 2/5$ and $q = 1/3$. The utilities of the two players (according to the true game parameters) in this distribution are $u_1 = 1/5$ and $u_2 = 1/3$ (for further details, see Appendix D). Surprisingly, these are the same utilities that our players obtained when telling the truth. In fact, this result is not a coincidence. The following theorem characterizes the equilibrium of the meta-game and the utilities obtained in it. While in our example we fixed a simple parameter space where $c$ was the single parameter of the first player and $d$ the single parameter of the second player, the following results apply to a wide range of parameter spaces where any one of the utilities of each player in the game is a parameter that the player can manipulate. As mentioned, this analysis is for the "limit" meta-game and holds for every pair of regret-minimizing agents.

**Theorem 4.** *In opposing-interests $2 \times 2$ games, the utilities of the two players in an equilibrium of the meta-game are the same as the utilities obtained when both play the truth in the meta-game.*

Thus, in these types of games the players do have incentives to unilaterally manipulate their own agents, but in equilibrium they can neither cooperate in the sense of improving their utilities nor do they suffer losses. This is in contrast to what we saw in dominance-solvable games and in Cournot competition games. Additionally, we characterize the cases where these games are manipulation-free:

**Theorem 5.** *An opposing interests $2 \times 2$ game is manipulation-free if and only if its Nash equilibrium is symmetric under player permutations (i.e., the same equilibrium distribution is obtained when player indices are switched).*

## 6 Further Related Work

Regret minimization in repeated strategic interactions and in online decision problems has been extensively studied in the literatures of game theory, machine learning, and optimization. Early regret-minimization algorithms were related to the notion of fictitious play [18, 79] (a.k.a. "follow the leader"), which in its basic form does not guarantee low regret, but its smoothed variants, such as "follow the perturbed leader" (FTPL) [41, 45, 50, 51] and "follow the regularized leader" (FTRL) [82], are known to guarantee an adversarial regret of $O(\sqrt{T})$ in $T$ decision periods; for more recent

---

can manipulate the parameter that he gives to his agent so as to indirectly cause an increase in $q$, the probability that the column agent plays the first column, an action that the row player finds favorable.

advances along these lines see, e.g., [27, 65, 88]. Other common approaches to regret-based learning are the "multiplicative-weights" algorithm, which has been developed and studied in many variants (see [1] and references therein, and see [23, 26, 77] for results on more advanced variants of this algorithm), and the family of algorithms that are based on the regret-matching approach that has been well studied in several settings (see [48] and references therein). For a broad discussion and for further references on regret-minimization dynamics, see [48] and [22].

Our work formalizes the meta-game faced by users of regret-minimizing learning algorithms, and asks whether and when users can benefit from manipulating their own agents. To our knowledge, there is no prior analysis or modeling of these strategic interactions between users of learning agents (which are induced by the dynamics of their agents). In a companion paper [53], we study an application of our meta-game model in auctions, where we analyze the dynamics and outcomes of repeated auctions played between regret-minimizing agents of a class that includes many natural algorithms such as multiplicative weights. We show in that paper that in the meta-game induced between the users of such auto-bidding agents, the second-price auction loses its incentive-compatibility property, while the first-price auction becomes incentive compatible.

As discussed in the introduction, our work is related to the broad field of equilibria in repeated games [61, 75], but the situation that the users of learning agents in our setting are facing differs from classic repeated games in significant aspects. Conceptually closer works are [16] that study equilibria between policies in the repeated game (i.e., strategies that are conditional on the history of play), and works on program equilibria [55, 73, 74, 89] in which each agent can read the commitments made by the other agents and condition its actions on these commitments. These models, however, are technically very different from our model, and in particular, the notions of equilibrium are different from a meta-game equilibrium.

In a broader perspective, our work is related to a research area that can be called "strategic considerations in machine-learning systems," with a growing body of work at the intersection of machine learning, algorithmic game theory, and artificial intelligence that addresses this topic from different perspectives, including learning from strategic data [19, 24, 32, 40, 43, 46, 58], Stackelberg games [12, 39], security games [38, 42, 69, 70, 84], and recommendation systems [10, 90]. More closely related works are [17, 30, 62], which deal with optimization against regret-minimizing agents. The possibility of obtaining increased gains when playing against a no-regret algorithm that is studied in these works is conceptually close to our work. A basic difference, however, is that these works consider a single optimizer facing a no-regret algorithm – a setting that induces optimization problems, rather than games – whereas we study games between the users that are induced by their learning agents' dynamics, and in which all users can act strategically and reason about the strategies of their peers. Additionally, in the meta-game a user does not need to select the actual step-by-step actions in the repeated game, but instead chooses the declaration to input into the automated agent, whereas the direct optimization over the action space of the underlying game is performed only by the agents.

Finally, the idea that inputting a "wrong" reward function into a learning algorithm can in some cases improve actual outcomes has been long known in the context of reinforcement learning [9, 54, 85, 86, 87]. Specifically, and closer to our context, in Markov games [60] it has been shown that the dynamics of reinforcement learners with certain intrinsic reward functions can lead to improved actual utility to all the agents playing the game [4, 33, 49, 57]. This literature, however, does not consider interactions between users of such agents and their incentives when entering their parameters into their learning agents. We view the analysis of our model for Markov games with reinforcement-learning agents as a natural and interesting extension, but we leave this for future work, and focus in the current paper on repeated games with regret-minimizing agents.

## 7 Conclusion

The present study deals with the modeling and analysis of scenarios in which human players use autonomous learning agents to perform strategic interactions with other players on their behalf. The usage of automated learning agents is becoming increasingly common and prominent in many real-world economic systems and online interactions such as online auctions [8, 21, 28, 29, 53, 68, 64, 72], financial markets [56, 59, 91, 93], and other systems [78]. Understanding the impact of this transition to automated agents on strategic systems and on their outcomes is a challenge in itself, and a prerequisite for studying how to better design such automated-interaction systems.

The framework that we propose for analyzing the meta-games between the users, and the results of our analysis, highlight the fact that the strategic nature of interactions does not disappear and does not remain the same when direct human play is replaced by automated agents, but rather the introduction of these learning agents changes in specific ways the rules of the game that the human users (who are the actual stakeholders in the system) are facing. Every automated agent that operates in a system on behalf of its user needs to receive some input from the user (such as preferences, goals, or constraints); we show that at this interaction point between a user and his own learning agent, the user faces a non-trivial strategic decision in which he needs to consider also the decisions of the other users. The goal of our model is to describe these interactions and formalize them.

Our contributions in this paper include the basic definitions related to these phenomena, demonstrations of how users of learning agents can profitably manipulate their own agents in several settings, as well as identification of cases where this cannot be done. These results have implications both for users of learning agents in strategic settings and for platform designers who need to take into account these types of manipulations. We believe that the present paper, along with its companion paper [53] where we study these phenomena in online auctions, only scratch the surface of these types of questions, and much work remains to be done.

## Acknowledgments

This project has received funding from the European Research Council (ERC) under the European Union's Horizon 2020 Research and Innovation Programme (grant agreement no. 740282).

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
