# OpenReview forum: "How and Why to Manipulate Your Own Agent: On the Incentives of Users of Learning Agents"
_NeurIPS.cc/2022/Conference — NeurIPS 2022 Accept_

### Official Review · Reviewer_Nu2E · 2022-07-07

**Rating:** 4
**Confidence:** 3
**Soundness:** 3 good
**Presentation:** 3 good
**Contribution:** 2 fair

**Summary:**

This paper investigates equilibria of a class of competitive games where players do not take actions directly, but instead pick a configuration for an (automated) learning agent which then engages in a multi-round competition with agents configured by the other players. The configuration is assumed to only affect the “declared utility” a player derives from each round’s outcome. The agents are assumed to employ a no-regret learning algorithm which maximises the *declared* utility. Under the authors’ assumptions, the studied games always have a coarse correlated equilibrium (CCE), implying the statistics of the repeated game _between the agents_ converge to the CCE in the infinite horizon limit. The authors then study the “meta-game” *between the players*, and their incentives to declare utility different from their underlying true one (which is assumed private).

The authors study three classes of games played by the agents: (i) dominance-solvable; (ii) (linear) Cournot competition; and (iii) opposing-interests games. In terms of “stability”, they find that players often have an incentive to declare utility different from their true one (“deceiving their agent”). Whether players achieve greater/lower utility in the CCE then depends on which of the above three agent games is played. The authors provide rigorous characterization of both (and say that deeper investigation of repeated auction is in a complementary paper, presumably also submitted to NeurIPS).

**Questions:**

1. Can you please provide real-world examples for where the setup you study exists? (See the “Strengths and weaknesses” review section.) If you can do so convincingly, I will change my recommendation to accept.

2. Can you please clarify why you are using the “single-shot” Nash equilibria as the baseline as opposed to some type of equilibria of the repeated game?

3. Is there ever an incentive to hire more than one SW agent? (In other words, how is the assumption of one agent per player justified?)

**Limitations:**

The crucial limitation of this work is in its motivation/applicability to realistic scenarios, as already discussed in other parts of the review. If there are none, it is hard to see merits; if there are, it is a good paper.

I did not look at the appendix which includes proofs.

**Strengths And Weaknesses:**

**Strengths**
+ With an increasing number of algorithms deployed in the real-world, scenarios where agents interact with each other are likely to increase in frequency. Understanding the resulting dynamics is therefore useful.

+ Breadth: The authors investigate a relatively large number of theoretical settings (dominance-solvable, Cournot, and opposing-interests games).

+ The authors fruitfully connect a variety of concepts from machine learning and economics.


**Weaknesses**
- AFAICT the authors do not provide a single reference to where a setup like theirs appears in the *real world*. The authors mention, e.g., auctions, but I am not aware of any instances where the *auctioneer* provides a _fixed_ (adversarial bandit) agent, with which the customers interact solely by declaring outcome utilities. IIUC, the assumption of an external fixed agent and utility declaration are crucial for relevance of these results, as more direct control over the agent would result in a standard repeated game (unless other constraints are introduced ofc). (Repeated games have ofc been studies extensively.)

- Relatedly, the paper explains technical definitions of the dominance-solvable/Cournot/opposing-interests games, but does not provide justification for why these particular games should be of interest (based on real-world applications).

---

> ### Author Response · Authors · 2022-08-02
> **Response to Reviewer Nu2E**
>
> We thank the reviewer for the feedback. We address below the main points that were raised in the review.
>
> 1.Motivation/applicability: As mentioned, one prominent motivating example for our setting is the case of online auctions with auto-bidders. Related relevant settings where our model can be applicable include, e.g., bilateral trade, exchange markets, and negotiation and bargaining algorithms. About auctions, the large internet companies provide in their ad-auction platforms auto-bidding agents for their users, and the users need to specify (in their user interface) parameters that define the utilities that the algorithmic agents aim to maximize. Some examples are the cost-cap and enhanced-CPC auto-bidders, where a desired maximum value for the average payment is set by the user, and the agent then optimizes w.r.t. this value, and a more complex example are budget-pacing algorithms that are widely used (see e.g., [1,2] below), which rely on budget reports from the users. The true values and budgets of the users are their private utility parameters, and it may well be that a meta-game analysis of these applications will show that users have incentives to misreport them. Such findings may have implications for design decisions in these platforms. The theoretical model that we propose for the meta game captures simple forms of auto-bidders, and of course we expect further work on more complex auto-bidders.
>
> As these auctions occur at fast rates (reaching thousands per second and more), the parameters that users set typically remain fixed for long sequences of auctions (e.g., budgets are often set monthly or weekly). We note that this is in fact inherent to the motivation of using automated agents – these agents are introduced in order to allow users not to constantly interact with the platform (e.g., either since they prefer not to do so, or they cannot effectively do so). In a model where users directly control the actions in the underlying game in a very frequent manner (not allowing the agents’ average utilities to stabilize via learning), the agents do not play this role, and this is not the setting that we consider.
>
> As for the classes of games studied: our focus is on modeling and demonstrating the types of phenomena that can happen due to strategic behavior of users of learning agents and on showcasing them in game classes that are as transparent for analysis as possible. Indeed we show that even in these simple settings, strategic play by the users gives rise to a rich variety of phenomena.
>
> Each of the three classes of games that we analyze has received an enormous amount of attention in previous literature. Specifically, as mentioned in Section 3, dominance-solvable games have applications in mechanism design and contract design. Cournot games are of course a classic economic model of competition that was studied in many papers, and our results have a clear economic interpretation. 2x2 games are a canonical setting that serves as an important testbed for new game-theoretic models such as the one that we propose, and specifically, learning dynamics in opposing-interests games (and in particular constant-sum ones) have been studied in many recent works. Therefore, we believe that these game classes are interesting and provide proper context and contrast to our results.
>
> Of course, we acknowledge that there is a tension between theoretical modeling and application, and real-world applications may present more complex challenges, and also that studying meta-games in specific applications can be extremely valuable. We believe that the model and type of analysis that we propose can both serve, as well as motivate, the study of meta-game phenomena that occur in application domains, and that insights from our results can carry over to such more complex settings.
>
>
>
> 2.Nash equilibria as a baseline: The motivation to compare outcomes where users act strategically to those of the Nash equilibrium of the underlying game, is that in the  game classes that we study this is the outcome that is obtained by regret-minimization dynamics in the case that users do not manipulate their parameter declarations. That is, the comparative static is between the case of strategic users (which is the case that we study) and the “null model” of non-strategic users.
>
>
>
> 3.One agent per user: There are cases where a platform would allow only a single identity per user, and we view the single-agent-per-user scenario as a natural starting point to begin our analysis of user incentives. We do agree that cases that allow users to deploy several sybil agents are of interest for further study and thank the reviewer for suggesting it.
>
>
>
> [1] Balseiro, S.R. and Gur, Y. Learning in repeated auctions with budgets: Regret minimization and equilibrium. (2019).
>
> [2] Gaitonde, J., Li, Y., Light, B., Lucier, B. and Slivkins, A. Budget Pacing in Repeated Auctions: Regret and Efficiency without Convergence. (2022).

---

### Official Review · Reviewer_WKoR · 2022-07-09

**Rating:** 7
**Confidence:** 4
**Soundness:** 3 good
**Presentation:** 4 excellent
**Contribution:** 3 good

**Summary:**

This paper introduces the study of how users interact with game-solving agents - in particular, it explores the question of what happens if we allow a user to specify, or indeed *mis*specify, her own utility function to a learning agent that will then attempt to learn and play an equilibrium on behalf of the user. The paper then establishes, somewhat counterintuitively, several cases in which the user has incentive to misspecify her utility, despite the fact that the learning agent's objective is perfectly aligned with the user's, and carefully analyzes the Nash equilibria of the resulting "meta-game" for the users in these various cases.

**Questions:**

1. Is there anything special about regret-minimizing agents in the context of this paper? It seems as though all the results in the paper hold if the learning agents are replaced with arbitrary black-box CCE (or Nash) solvers. If this is the case, I think it'd be better to frame the result in this light, as it would be more general.
2. To what extent is it possible to "mechanism design" around this problem? For example, can one program the learning agents in such a way that the users are incentivized to tell the truth? At what cost does this come? For example, such learning agents can't always converge to CCE, lest they be subject to the incentive problems raised by this paper, and therefore they cannot all be regret minimizers. Can we say anything positive?
3. Thm 4 reminds me of the revenue equivalence principle in mechanism design. Is there any connection here?

**Limitations:**

Yes.

**Strengths And Weaknesses:**

This paper is very intellectually stimulating and well written, and I think it brings to light an important and quite counterintuitive concept. For that, I think it is worthy of acceptance.

If I can point out any weakness, it is perhaps the specificity of the theoretical results: they are limited to special cases of dominance-solvable, Cournot, or 2x2 games. I do not think that this weakness is significant though--it simply leaves room for better future work!

I also think that one of the points raised only in the appendix--namely, that regret minimization alone is never guaranteed to converge to any particular CCE in a game with multiple CCEs--is interesting in its own right, and perhaps deserves a mention in the main body.

---

> ### Author Response · Authors · 2022-08-02
> **Response to Reviewer WKoR**
>
> We thank the reviewer for the feedback. We address below the main points that were raised in the review.
>
> 1.Regret minimization: It is true that if you replace the regret-minimization dynamics with a black-box solver that solves for a CCE with the declared parameters, then all our results still hold. We find this view, however, less natural and suitable in our setting and prefer using regret-minimizing agents for two main reasons. The first reason is that of motivation: no regret is of course a general and widely used notion of learning outcomes, and there exist many practical algorithms that have the regret-minimization property and are suitable for our setting, while we find black-box solvers to be less well motivated.
>
> The second reason is due to the information structure: to obtain our results on meta-games, the solver will need to be able to calculate the CCE for any set of parameter declarations that the agents get. While regret-minimization dynamics manage to “calculate” the CCE in an uncoupled manner (where every agent gets input only from its own user and is aware only of its own utility), a solver will need to receive as input all the parameter declarations from all the users. Thus our results extend to black-box solvers, but in a model where the parameters of the agents are no longer private, or that agents (truthfully) communicate their utilities to the solver. We view the information structure that we consider in which the parameters that an agent gets remain private as a more natural model for our setting where agents optimize on behalf of their users. We will add a comment about this in the paper.
>
>
> 2.Mechanism design: Mechanism design in meta-game settings is an interesting question that we have not explored yet. We agree that designing algorithms that, at least for given classes of games, will reach desirable outcomes in the meta-game via their dynamics (such as inducing truthfulness as suggested, maximizing welfare, etc.) is an interesting direction to study. An alternative approach would be to not restrict ourselves to only algorithm design and to study settings where the designer has even more power and is able to design both the game and the agents, possibly with some requirements on both (this would be quite natural in online auction platforms where the platform runs the auction and also provides automated bidding agents for its users).
>
>
> 3.Relation to the revenue-equivalence theorem: We agree that our result in Theorem 4 may indeed look similar to the result of the revenue equivalence theorem, but we do not know of a formal connection between the two results.

---

> > ### Comment · Reviewer_WKoR · 2022-08-06
> > **Response to response**
> >
> > Thank you for the response. I thought this was a good paper before, and I still do. I do not change my score.

---

### Official Review · Reviewer_Uqbg · 2022-07-09

**Rating:** 6
**Confidence:** 4
**Soundness:** 3 good
**Presentation:** 3 good
**Contribution:** 3 good

**Summary:**

The authors consider a setting where human players play a game, but automated agents execute their decisions. This scenario arises in, e.g., online advertising, where an automated bidding agent makes real-time decisions intending to maximize the utility of the player it represents. The authors distinguish between the meta-game between human players and the game instantiation the agents play. The instantiation is induced by players specifying parameters like their utility for outcomes (valuations, costs, etc.). The authors claim that automated agents are regret-minimizing algorithms in many settings.

The authors identify a loophole in this modeling: Aiming to maximize their utility, the players might be incentivized to manipulate their agents. Namely, there are cases in which passing the wrong parameters to agents can actually lead to better player utility.

The authors analyze three classes of games and show when/how manipulation can help players. They provide elaborate examples along with the formal characterization of beneficial manipulations.

**Questions:**

Overall, I'm positive about this paper but want to understand the author's stance about the weaknesses I raised. Please address my 1 and 2 concerns in the "weaknesses" part.

**Limitations:**

Null

**Strengths And Weaknesses:**

Strengths:

1. The topic is hot and relevant. Real-world applications that employ automated agents (like online advertising) are positioned in multi-billion industries. Even small advancements can be highly profitable, so reasoning about potential harms is highly impactful.

2. The claims are non-intuitive. Regret-minimizing algorithms are used widely, and the authors prove they can become problematic in multi-agent settings. Furthermore, the authors provide deep insights like the connection between regret-minimizing to Stackelberg games and cooperation in Theorem 1.

3. The paper is full of examples that assist in understanding the different classes of games, the deviations, and the theoretical results.

Weaknesses:
1. My main concern about this paper is the so-called companion paper (first presented in line 80). The authors, it seems, already initiated the study of "incentives of users in automated learning agents in online strategic systems" with that companion paper (assuming it was published). Hence, line 67 is misleading: only one of those papers could initiate this study. Further, I'm not familiar with treating previous literature as a "companion paper" --- how should I assess the contribution of this paper without knowledge about the other? I believe the authors should have addressed their previous "companion" paper as related work in this paper. I understand that addressing the companion paper risks anonymity, and yet the current situation where there's another paper on this topic, but it is not surveyed here is also unfortunate.

2. If I understand correctly, the incentive misalignment follows from the automated agents being regret minimizers. Therefore, I was surprised to see that the authors don't formally define the notion of regret. Further, they discuss no-regret dynamics or other essential notions such as CCE without defining them. I believe that defining these technical notions is crucial for understanding the paper's contribution (and appreciating the paper's non-intuitive claims about the incentive mismatch).

3. I did not consider the following comment in my score, but I believe the authors could significantly improve the write-up. Here are several examples:
* The order of citations is incorrect (e.g., [75, 43, 5] in line 127, instead of [5, 43, 75])
* Many sentences are unclear. For instance, line 143 says, "this simple analysis also shows that the other player has..." Who is the other player? The word "this" at the beginning of the sentence is a dangling modifier. Many other sentences were hard to follow.
* Inconsistency of technical details, for instance, "sub-family … [of] games" vs. "subclass of… games" in lines 262 and 264. Such inconsistencies make reading harder.
* Wordiness, for example, line 222: "Analyzing the equilibrium of the meta-game, one obtains that the equilibrium of the meta-game." Why should we repeat "equilibrium of the meta-game" twice? I believe that the authors could be more articulate in this sentence.

---

> ### Author Response · Authors · 2022-08-02
> **Response to Reviewer Uqbg**
>
> We thank the reviewer for the feedback. We address below the main points that were raised in the review.
>
> 1.Companion paper: We would like to note that the format of companion papers is not new and appears in the literature (for some examples of companion-paper pairs, see, e.g., references [1-6] below). We found this format most appropriate in our case since the two studies were done and written concurrently and complement each other.
>
> We wish to emphasize and clarify that there is a clear separation between the contributions of these two papers. The paper on auctions only focuses on auctions and technically focuses on convergence and equilibrium selection questions that become interesting in the auction setting. Importantly, the paper on auctions clearly gives credit to and cites the current paper as the one that formalizes and studies the meta-game model, and it does not claim to initiate the study of incentives of users of automated learning agents.
>
> We understand that not providing the reference is not optimal, but since the papers are indeed companion papers, we were careful not to provide the citation in order not to jeopardize the anonymity requirement of the submission.
>
>
> 2.Definitions of regret and coarse correlated equilibria: We agree that the technical definitions of regret and coarse correlated equilibria are important for clarity and completeness. We note that the definitions that we consider are standard and appear in books that we cite, and so we allowed ourselves to postpone the definition of the regret-minimization property to Appendix 1. We thank the reviewer for pointing this out and will add a reference to the definitions from the main text, and we will also add the formal definition of coarse correlated equilibria.
>
>
> 3.We thank the reviewer also for the feedback on the write-up. We will use it to further improve the paper.
>
> ------
> [1] Roughgarden, T. and Tardos, É., 2002. How bad is selfish routing? (Journal of the ACM)
>
> [2] Roughgarden, T. and Tardos, É., 2002. Bounding the Inefficiency of Equilibria in Nonatomic Congestion Games. (Later journal publication in Games and Economic Behavior 2004)
>
> ------
> [3] Karp, R.M., Upfal, E. and Wigderson, A., 1986. Constructing a perfect matching is in random NC. (Combinatorica)
>
> [4] Karp, R.M., Upfal, E. and Wigderson, A., 1985, December. Are search and decision programs computationally equivalent? (STOC 1985)
>
> ------
> [5] Raghu, M., Poole, B., Kleinberg, J., Ganguli, S. and Sohl-Dickstein, J., 2017, July. On the expressive power of deep neural networks. (ICML 2017)
>
> [6] Poole, B., Lahiri, S., Raghu, M., Sohl-Dickstein, J. and Ganguli, S., 2016. Exponential expressivity in deep neural networks through transient chaos. (NeuIPS 2016)

---

> > ### Comment · Reviewer_Uqbg · 2022-08-07
> > **Response**
> >
> > Thanks for your reply.

---

### Official Review · Reviewer_zZ5S · 2022-07-20

**Rating:** 6
**Confidence:** 3
**Soundness:** 4 excellent
**Presentation:** 4 excellent
**Contribution:** 3 good

**Summary:**

The paper suggests to view the strategic behaviour of _users_ of agent representatives, assuming those representatives are playing a repeated normal form game using regret-minimisation. As opposed to many papers that would present a limited set of specific examples, authors present analysis of game _classes_ where users are an incentive to mis-represent their actual preferences to manipulate the interaction between the automated representatives.

**Questions:**

There are authors who study out-of-equilibrium behaviour and "stable cycles". Are there any indications of what happens if the automated agents are locked into cycle/dynamic-equilibrium patterns, rather than strongly converge to a (pure) equilibrium in beliefs/traces?

Along this same line of thinking. Since the model already admits separation of representative and user utility, what happens if the meta-game utility is even further removed. E.g., trading on "futures", deriving utility from the representative agent beliefs, rather than decisions that those beliefs produce in the underlying game at that moment?


**Limitations:**

none beyond the questions above


**Strengths And Weaknesses:**

Strength: Analysis of manipulation of adaptive learning _systems_. Applicable (potentially) to multi-agent learning scenarios, especially hierarchical learning approaches. Game _class_ analysis.

Weaknesses: Assumption of an equilibria seeking behaviour. Needs at least a mention of off-equilibria behaviour possibility (see works by

---

> ### Author Response · Authors · 2022-08-02
> **Response to Reviewer zZ5S**
>
> We thank the reviewer for the feedback. We address below the main points that were raised in the review.
>
> 1.Out-of-equilibrium patterns and dynamic equilibria: We would like to emphasize that the notion of convergence that is considered in our analysis is only convergence of the average empirical play (a.k.a. average-iterate convergence), and not the stronger convergence of the strategies of play to a fixed strategy profile (a.k.a. last-iterate convergence). This is so since the (joint) average empirical play of the agents is the thing that determines the long-term payoffs of the users, and so whether or not last-iterate convergence occurs does not play a role in our analysis, and the equilibrium that is obtained can certainly be a dynamic one.
>
> In all the game classes we analyze, average-iterate convergence holds for any regret-minimization dynamics, even if the dynamic of the strategies itself reaches stable cycles or other notions of recurrence (e.g., as in [1] below). While for some games and algorithms the last-iterate converges as well, in other cases it does not. For example, in opposing-interests games, for many standard regret-minimization algorithms the dynamics of strategies of play indeed cycle away from the Nash equilibrium profile, while at the same time the long-term empirical average does converge (see Section 5 and Appendix 4 and references 6, 7 in the paper). Additionally, in Appendix 1 we show and discuss cases where regret-minimization dynamics do not converge even in the average-iterate sense.
>
>
> 2.Other utility models: The suggestion of analyzing situations where the user utility is even further removed from the agent utility (or considering alternative models of strategic reasoning, such as ,e.g., level-k models) indeed sounds like an interesting direction for further research, but we have not studied it.
>
>
> [1] Mertikopoulos, Panayotis, Christos Papadimitriou, and Georgios Piliouras. "Cycles in adversarial regularized learning." (2018).

---

### Meta-Review · Area_Chair_xBXo · 2022-08-24

**Recommendation:** Accept
**Confidence:** Less certain

**Metareview:**

Most reviews are positive and think that the paper solves an interesting and non-trivial problem. One reviewer points out some concerns on the motivating example, and it seems to be addressed in the author response. I have a different concern that in real world, the set of eligible bidders in each auction differs a lot, and maybe the authors can add some discussion on how this affects the result.

**Award:**

No

---

### Decision · Program_Chairs · 2022-09-14

Accept